# Combined Fluorescence-Guided Resection and Intracavitary Thermotherapy with Superparamagnetic Iron-Oxide Nanoparticles for Recurrent High-Grade Glioma: Case Series with Emphasis on Complication Management

**DOI:** 10.3390/cancers14030541

**Published:** 2022-01-21

**Authors:** Michael Schwake, Michael Müther, Ann-Katrin Bruns, Bastian Zinnhardt, Nils Warneke, Markus Holling, Stephanie Schipmann, Benjamin Brokinkel, Johannes Wölfer, Walter Stummer, Oliver Grauer

**Affiliations:** 1Department of Neurosurgery, University Hospital Münster, 48149 Münster, Germany; michael.muether@ukmuenster.de (M.M.); ann-katrin.bruns@ukmuenster.de (A.-K.B.); nils.warneke@ukmuenster.de (N.W.); markus.holling@ukmuenster.de (M.H.); stephanie.schipmann@ukmuenster.de (S.S.); benjamin.brokinkel@ukuenster.de (B.B.); walter.stummer@ukmuenster.de (W.S.); 2European Institute for Molecular Imaging (EIMI), University of Münster, 48149 Münster, Germany; bastian.zinnhardt@ukmuenster.de; 3Biomarkers & Translational Technologies (BTT), Pharma Research & Early Development (pRED), F. Hoffmann-La Roche Ltd., CH-4070 Basel, Switzerland; 4Department of Neurosurgery, Haukeland University Hospital, 5021 Bergen, Norway; 5Deparment of Neurosurgery, Hufeland Klinikum, 99974 Mühlhausen, Germany; johannes.woelfer@t-online.de; 6Department of Neurology with Institute of Translational Neurology, University Hospital Münster, 48149 Münster, Germany; oliver.grauer@ukmuenster.de

**Keywords:** glioma, myeloid cells, nanoparticles, thermotherapy, fluorescence, inflammation

## Abstract

**Simple Summary:**

Recurrent high-grade gliomas are difficult to treat. Here, we report on our single-center experience in combining fluorescence-guided tumor resection with 5-ALA and local thermotherapy with superparamagnetic iron nanoparticles. In total, 18 patients were operated on and received thermotherapy with or without additional radiotherapy. The median progression-free survival was 5.5 months and median overall survival was 9.5 months. Although no major side effects were observed during active treatment, 72% of the patients developed cerebral edema requiring steroid treatment or even surgical removal of the nanoparticles. In conclusion, the combination of fluorescence-guided resection and intracavitary thermotherapy provides a novel and promising treatment option for improving local tumor control in recurrent high-grade gliomas, but further refinements of the treatment protocol are needed to decrease major side effects.

**Abstract:**

Background: Concepts improving local tumor control in high-grade glioma (HGG) are desperately needed. The aim of this study is to report an extended series of cases treated with a combination of 5-ALA-fluorescence-guided resection (FGR) and intracavitary thermotherapy with superparamagnetic iron oxide nanoparticles (SPION). Methods: We conducted a single-center retrospective review of all recurrent HGG treated with FGR and intracavitary thermotherapy (*n* = 18). Patients underwent six hyperthermia sessions in an alternating magnetic field and received additional adjuvant therapies on a case-by-case basis. Results: Nine patients were treated for first tumor recurrence; all other patients had suffered at least two recurrences. Nine patients received combined radiotherapy and thermotherapy. The median progression-free survival was 5.5 (95% CI: 4.67–6.13) months and median overall survival was 9.5 (95% CI: 7.12–11.79) months. No major side effects were observed during active treatment. Thirteen patients (72%) developed cerebral edema and more clinical symptoms during follow-up and were initially treated with dexamethasone. Six (33%) of these patients underwent surgical removal of nanoparticles due to refractory edema. Conclusions: The combination of FGR and intracavitary thermotherapy with SPION provides a new treatment option for improving local tumor control in recurrent HGG. The development of cerebral edema is a major issue requiring further refinements of the treatment protocol.

## 1. Introduction

In spite of guideline-adherent treatment, the prognosis of high-grade gliomas (HGG) is poor [1,2]. Over time, developments such as fluorescence-guided resection (FGR) [3,4], intraoperative neuro-monitoring (IOM) [5,6], intraoperative magnetic resonance imaging (ioMRI) [7,8] and novel treatment methods such as Tumor-Treating Fields (TTF) [9] have led to limited improvements of survival times. 

Fluorescence-guided resection with 5-aminolevulinic acid (5-ALA) is currently the only approved and one of the most widely used procedures in adult glioma surgery. The efficacy of 5-ALA-assisted resection and its impact on the degree of resection, progression-free survival (PFS), and overall survival (OS) in primary cases have been demonstrated in numerous studies [3,4,5,10,11]. Stummer et al. identified a significantly higher OS when no residual tumor was visible after surgery (16.7 vs. 11.8 months, *p* < 0.0001) [12]. Moreover, this method is based on the intracellular accumulation of fluorescent protoporphyrin IX (PpIX). This selective accumulation is the reason for the high sensitivity and specificity of the fluorescence signal [13,14,15,16,17,18], also in the case of recurrent malignant brain tumors [19]. 

Therapies for tumor recurrence are not well defined, and the overall level of evidence for available treatments is low. Current guideline recommendations include re-resection, re-irradiation, chemotherapy and experimental treatment options [20]. In recent years, several clinical trials have yielded only limited results [21,22]. Even though HGG affect the whole brain, most surgical cases recur locally at the margins of resection cavities [23,24]. Therefore, improving local tumor control is the aim of all local therapy strategies, such as locally applied chemotherapy [25] and photodynamic therapy (PDT) [23,26].

Early clinical trials demonstrated the effectiveness and safety of stereotactic application of superparamagnetic iron oxide nanoparticles (SPION) subjected to an alternating magnetic field (AMF) in combination with irradiation in patients with recurrent HGG [27,28]. SPION with AMF is an approved treatment within the European Union. Hyperthermia generated by SPION has been shown to possibly elicit a potent antitumor immune response [29,30,31] with enhanced infiltration of natural killer (NK) cells, macrophages, dendritic cells and CD4^+^ and CD8^+^ T cells [32]. 

One major technical issue with stereotactic application is the inadequate distribution of SPION [28]. Additionally, with stereotactic procedures, immediate cytoreductive treatment is not possible. With this in mind, we developed a novel technique of administering the particles into the resection cavity immediately after FGR. The main advantage of this method is that the surgeon has optimal control over particle distribution, enabling him to apply larger particle volumes while preventing spillage into the ventricles. 

The primary aim of this study is to present our results concerning effectiveness and the outcome, in term of overall survival (OS), with the secondary objective to evaluate the safety of combined FGR and local intracavitary thermotherapy with superparamagnetic iron oxide nanoparticles in an extended series of patients treated at our center. 

## 2. Methods

### 2.1. Patients

This case series includes all patients treated with a combination of FGR and intracavitary SPION thermotherapy in a single-center academic setting from 2015 to 2020 (Figure 1). All patients with recurrent HGG considered as candidates for surgical resection were offered additional intracavitary thermotherapy as an adjunctive treatment modality. Data analysis was performed retrospectively, with all procedures and analyses being approved by the local ethics committee according to the declaration of Helsinki (2020-531-f-S). Informed consent was obtained from all subjects involved in the study.

### 2.2. Nanoparticles

As previously reported [32], the semifluid solution MFL AS-1 (NanoTherm^®^, MagForce AG, Berlin, Germany) containing SPION with an iron concentration of 112 mg/mL was applied during surgery. The fluid is manufactured according to European medical device regulations and has been approved for the treatment of brain tumors since 2011. 

### 2.3. Fluorescence-Guided Resection and Nanoparticle Application

Tumor resection was performed using a standard microsurgical technique. 5-aminolevulinic acid (5-ALA, Gliolan^®^, Medac GmbH, Wedel, Germany) was applied orally 4 h prior to anesthesia induction at a concentration of 20 mg/kg body weight. The surgical aim was to remove the fluorescent tumor. In cases with a motor-eloquent tumor location, dynamic MEP mapping was used for corticospinal tract mapping. Speech-eloquent tumors were resected under awake conditions for speech monitoring, as described previously [33]. After tumor resection, the cavity wall was coated with a hydroxycellulose mesh (Tabotamp^®^, Johnson & Johnson Medical GmbH, Ethicon, Norderstedt, Germany), and subsequently SPION were applied to the hydroxycellulose mesh using a 1 mL syringe. Up to three layers of hydroxycellulose and SPION were applied, if possible. Additionally, a closed-end catheter (TK-01, MagForce AG, Berlin, Germany) was led through the particle deposits to allow temperature measurement during treatment in the AMF device (Nano-Activator^®^, MagForce AG, Berlin, Germany). The bone flap was re-fixed using absorbable, non-metal plates (SonicWeld Rx^®^, KLS Martin, Tuttlingen, Germany). Metallic body implants up to 40 cm from the tumor, including dental implants and bone flap fixation plates, were removed during surgery or were removed before. Patients in which tumor resection led to opening of the ventricle were not eligible for the application of nanoparticles to decrease the risk of unintended particle distribution within the CSF spaces.

Due to artifacts caused by the SPION, we could not rely on postoperative MRI scans when estimating the extent of resection (EOR). Therefore, we rated EOR as gross total resection (GTR) if all fluorescing tumor tissue was removed, near total resection (NTR) whenever a weak and diffuse fluorescent signal was detected at the end of surgery, and subtotal resection (STR) if a strong and compact fluorescence signal had to be left over, e.g., in eloquent regions. Previous studies have proven a close association between the fluorescence signal and EOR on postoperative MRI [3,5,12].

### 2.4. Thermotherapy

After surgery, computed tomography (CT) was performed (Figure 2). These images were fused with preoperative MRI scans using the treatment simulation software NanoPlan^®^ (MagForce AG), as published previously [32,34,35,36]. NanoPlan^®^ simulates heat generation as a function of the nanoparticle density and intensity of AMF. However, due to the lack of technical possibilities for measuring local tissue perfusion over the course of a hyperthermia application, this simulation tends to be imprecise. Therefore, temperature measurements must be performed during the treatment session to readjust the simulation. During the first hyperthermia application, a fiber-optic temperature sensor (Optocon, Dresden, Germany) was inserted into the closed-end catheter that was placed during surgery. Thermotherapy was performed in the AMF applicator operating at a frequency of 100 kHz and with field intensities of 2.5–15 kA/m, as published previously [32]. The highest temperature along the thermocatheter course was used to fine-tune the field strength. Two one-hour treatments were scheduled per week. In cases with concomitant radiotherapy, the first thermotherapy session was scheduled 3 days before the start of radiotherapy, while another five sessions were conducted at days 1, 4, 8, 11, 15 ± 1 day. Radiotherapy was performed as previously described [32] and, when planned for the same day, took place within a time interval of 2 h to each thermotherapy session.

### 2.5. Follow Up

As noted above, the placement of iron nanoparticles does not allow for adequate postoperative imaging with MRI. Patients who live close to our center were followed up with ^18^F-Fluoro-ethyl-tyrosine (^18^F-FET)-PET-CT, including post-contrast studies, once every three months. With all other patients, post-contrast CT scans were used for follow-up. If the nanoparticles had to be removed over the course of disease, MRI was again chosen as the optimal imaging modality. In order to further visualize the immunological response during thermotherapy, we conducted a *N*,*N*-diethyl-2-[4-(2-fluoroethoxy)phenyl]-5,7-dimethylpyrazolo[1,5-a]pyrimidine-3-acetamide (^18^F-DPA-714) PET scan for translocator protein (TSPO) in individual cases, as previously described [37].

### 2.6. Data Analyses

The IBM SPSS Statistics 25.0 package (IBM, Armonk, New York, NY, USA) was used for statistical analyses. Data were analyzed by standard descriptive statistics, using absolute and relative frequencies for categorical variables, median and interquartile range (IQR) for continuous variables, and mean and standard deviation (SD) for metrical variables. The Mann–Whitney U-Test (MWU) was used for ordinal and Fisher’s exact test for categorical variables. Time-to-event analyses were performed using Kaplan–Meier curves and log-rank test. Progression-free survival (PFS) defines the time from the procedure until progression according to modified RANO criteria or death [38]. Overall survival (OS) was defined as the time from procedure to death. A probability value less than 0.05 was considered statistically significant. 

## 3. Results

Eighteen patients with recurrent HGG were treated with FGR and subsequent intracavitary thermotherapy. Eight patients (44%) were female. The median age was 51 years (IQR: 43–61). According to the WHO classification of 2006 [39], sixteen patients (89%) were diagnosed with glioblastoma, and each of the remaining two were diagnosed with anaplastic astrocytoma and anaplastic oligodendroglioma, respectively. Fourteen cases showed IDH wildtype (78%). The MGMT promotor was non-methylated in 13 cases (76%). Nine patients (50%) were treated for first tumor recurrence; all other patients suffered at least two recurrences (Table 1). The median time between initial diagnosis and surgical treatment investigated in this study was 14 months (IQR: 7–45). All patients received thermotherapy twice a week at a median temperature of 47.0 °C (IQR: 44.5–53.3). Nine patients (50%) additionally received concurrent radiotherapy at a dose of 39.6 Gy (5 × 1.8 Gy/week); all other patients did not receive radiotherapy due to dose limitations *(*Figure 1*)*. Initially, no salvage chemotherapy was administered. During follow-up, one patient received temozolomide and one patient received bevacizumab for palliative edema treatment *(*Table 2). 

### 3.1. Survival Analysis

Tumor progression was defined according to the modified RANO criteria [38]. The median progression-free survival (PFS) for the study population was 5.5 months (95% CI: 4.67–6.13) after thermotherapy, and median overall survival (OS) was 9.5 months (95% CI: 7.12–11.79) (Figure 3). Survival differences could neither be observed between patients treated for first recurrence and patients treated for second recurrence or later (*p* = 0.283 for PFS; *p* = 0.608 for OS) nor in patients who received both thermotherapy and radiotherapy and those having received thermotherapy alone (*p* = 0.232 for PFS; *p* = 0.450 for OS). 

### 3.2. Safety and Complications 

No major side effects were observed in the immediate perioperative phase or during AMF treatment. However, thirteen patients (72%) developed cerebral edema with clinical symptoms during treatment follow-up (median: 70 days, IQR: 50–105) and were treated with dexamethasone. In six of these thirteen cases (46%), nanoparticles had to be removed surgically due to refractory edema. Four (31%) of these developed impaired surgical site infections after prolonged treatment with steroids (median: 72 days, IQR: 59–116). One patient (No. 4) developed transient mild myelopathy, with cervical MRI suggesting a T2 hyperintense medullary heat injury caused by a small spinal deposit of dislocated nanoparticles (Figure 4).

The mean peak temperature during treatments was 47 °C (IQR: 44.5–53.3). We found no significant association between peak temperature and edema development (*p* > 0.05). Moreover, no significant correlations between peak temperature and steroid treatment, reoperation, OS, PFS, or time interval to edema development (all *p* > 0.05) (all *p* > 0.05) were found. Further analyses did not show any association between additional radiotherapy and the development of edema (*p* > 0.05). 

## 4. Discussion

In this analysis of an extended case series, we evaluated the effectiveness, safety and technical issues of combined 5-ALA FGR and intracavitary thermotherapy. We found a median PFS of 5.5 months (95% CI: 4.67–6.13) and median OS of 9.5 months (95% CI: 7.12–11.79) after surgery and thermotherapy. In contrast to earlier studies, we did not observe a significant value of re-irradiation in combination with thermotherapy on OS and PFS [32,40,41,42]. Moreover, we found no significant difference between the survival outcomes of patients treated for their first recurrence vs. second recurrence or later. 

As for now, there is no standard treatment for patients with recurrent HGG. Recent non-randomized trials show that patients may benefit from repeated resection with a median PFS of 1.9 months and OS of 6.5–12.9 months [43,44,45,46]. Re-irradiation was also associated with a modest survival benefit compared to best supportive care alone. Survival times could be improved when re-irradiation was combined with other treatment modalities (median OS of 8.2 months vs. 12.2 months) [47]. Regarding systemic treatments in recurrent GBM, therapeutic options include salvage chemotherapy with temozolomide, lomustine, bevacizumab or combinations, and regorafenib. A median PFS of 1.5–4.2 months and median OS of 6.0 to 10.6 months were reported [48,49,50,51,52]. Decision making is even more complex in cases of second or later recurrence. In general, thermotherapy is not a novel concept in neuro-oncology. Laser interstitial therapy (LITT), a stereotactic procedure which is limited to smaller lesions, showed comparable survival outcomes for recurrent glioblastoma [53,54]. Recently, our group also published a series of recurrent glioblastoma patients treated with intraoperative open photodynamic therapy after FGR with comparable survival curves [23]. Importantly, in comparison to other modalities, NanoTherm^®^ nanoparticles have been certified according to the European medical device regulations for the treatment of brain tumors since 2011. Moreover, intracavitary nanoparticle application allows one to address more complex tumor architecture and can be directly combined with microsurgical resection.

### 4.1. Combining FGR with Application of Nanoparticles

In comparison to the classical stereotactic application of nanoparticles [27], its addition to fluorescence-guided resection has several advantages. Cytoreduction using FGR and IOM shows good oncological and functional results [55]. This allows us to provide thermotherapy in addition to the best standard of care. As opposed to stereotactic application of nanoparticles, higher volumes of nanoparticles can be applied without causing elevated intracranial pressure. Moreover, as an open procedure enables the neurosurgeon to distribute the particles more accurately under direct visual control, common problems connected to the stereotactic method including leakage and backwash of the nanoparticles, which can be avoided. Finally, with one surgical corridor, there is no need for multiple trajectories, which are required in cases of larger tumors with stereotactic application alone [27,28]. 

### 4.2. Complication Management

After a short period of time (median: 70 days, IQR: 50–105) more than half of the patients developed perifocal edema with additional neurological deficits, requiring treatment with higher steroid doses. In six of these cases, nanoparticles had to be removed surgically due to refractory edema (*n* = 6; 33%). In our previous publication we focused on the immunological aspect of this reaction that must be considered as part of the treatment. In histopathology, brain tissue revealed large amounts of aggregated nanoparticles located in necrotic tissue without evidence of tumor activity. At the borders, nanoparticles were found to be incorporated by phagocytes. The surrounding tissue exhibited a strongly proinflammatory state with increased T cell, NK cell and myeloid cell infiltration [32]. These reactions were also found in this case series (patients 9 and 13). As well as edema formation, four patients also developed surgical site infection, probably due the prolonged treatment with steroids (*n* = 4; 22%). Therefore, the administration of high doses of corticosteroids (dexamethasone more than 4 mg/day) for a longer period (>6 weeks) should be avoided. As an alternative, bevacizumab could be used for edema control, as has been demonstrated for the treatment of radionecrosis [56].

Otherwise, the surgical removal of the nanoparticles and of necrotic tissue bulks from the tumor core must be discussed. In addition, thermotherapy should be limited to patients who are not in need of corticosteroids before treatment starts. Moreover, patients with compromised clinical performance status who may not tolerate temporary increases in edema secondary to necrosis, or are in risk of other major side effects, will not be eligible for this therapy—as is also the case for all other therapies.

In comparison to stereotactic applications [27,28], nanoparticle application after tumor resection allows for a much better distribution and higher concentration of nanoparticles at the borders of the tumor cavity. Yet, elution into the ventricles and basal cisterns must be avoided to prevent complications such as in patient 4. Therefore, any contacts between the resection cavity and ventricles or cisterns should be sealed, e.g., using fibrinogen-coated collagen pads, if possible.

### 4.3. Outlook

As hyperthermia must be assumed to be the major cause for edema formation, special attention must be directed to temperature simulation and adjustment. Currently, the maximum temperature registered along the course of the closed-end catheter is used to correct the simulation and to tune the AMF. It might be more reasonable to adjust the AMF according to the temperature at the tumor–SPION border, for this is the region in which biological heat effects are expected to be the most pronounced. 

We are working on solutions to measure temperatures simultaneously at multiple sites along the thermometry catheter, which should preferably be positioned along the rim of the tumor cavity. These measurements will be used for real-time corrections of the temperature simulation to create a more precise temperature chart. Another option might be to place a second catheter to obtain more reference points for the simulation. It is not easy to envision, but most the desirable process for accurate therapy steering would be a non-invasive method able to create a three-dimensional temperature chart under the conditions of a high-energy alternating magnetic field.

The main determinants of tissue heat generation are field strength, nanoparticle density and tissue perfusion. A standard perfusion value averaged over the whole brain is currently used for temperature simulation. Because of postoperative scarring, necrosis, varying cell densities or preexisting edema, etc., local perfusion will almost always differ from this generalized assumption. As brain perfusion mapping can be easily and routinely accomplished using dynamic contrast CT techniques, the NanoPlan^®^ software is currently modified to include this modality for a voxel-based correction of the simulation. 

A higher number of nanoparticle sheets allows for lower AMF energy, which facilitates field strength adjustments during treatment sessions. However, too large volumes may foster space-occupying necrotic reactions with post-treatment edema, as seen in case number 13 (Table 2). More highly concentrated SPION solutions might be engineered to address this problem.

One novel option to demonstrate and monitor post-treatment inflammatory reactions and the immunologic tumor microenvironment (TME) is dual-tracer PET imaging with ^18^F-FET and ^18^F-DPA-714 (a ligand of microglial translocator protein (TSPO)). In the case of patient 8, this imaging was conducted six months after thermotherapy (Figure 5). TSPO signaling clearly exceeded the FET signal, which might indicate the activation of glioma-associated myeloid cells beyond the tumor borders delineated by FET-PET. Combining these imaging biomarkers supports the characterization of the TME, as discussed in an earlier study [37]. Regarding thermotherapy, TSPO imaging shows perifocal immunologic reactions that clearly exceed solid tumor volumes. Therefore, TSPO PET may contribute to a more elaborate classification of recurrence that helps the clinician to perfectly tailor the thermotherapy settings [57].

At present, we assume that FGR and intracavitary thermotherapy are a good alternative treatment option for patients with recurrent and resectable HGG in non-eloquent regions with moderate edema. This particularly includes HGG patients at first recurrence with a non-methylated MGMT promotor and tumor progression during the treatment with alkylating agents, but also HGG patients with second or later recurrence. However, our results are based on a retrospective review of a heterogenous series of recurrent HGG. Further investigations are necessary to refine several technical aspects, such as the applied SPION volume, or the calculation and monitoring of therapy temperature. Recently, a patient registry has been established to collect treatment data prospectively.

## 5. Conclusions

FGR in combination with intracavitary thermotherapy is an interesting treatment modality for patients with malignant glioma. In addition to cytoreductive treatment, non-ablative hyperthermia can induce an inflammatory reaction; however, the frequent development of cerebral edema requires the refinement of the treatment protocol.

## Figures and Tables

**Figure 1 cancers-14-00541-f001:**
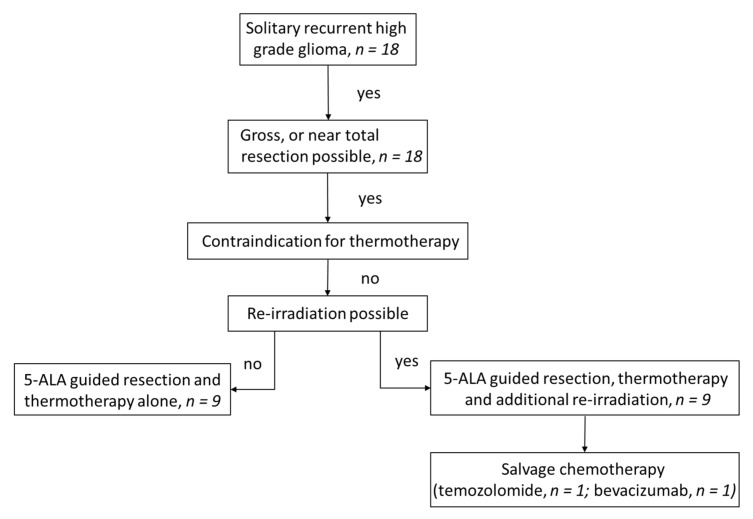
Flow chart of treatment.

**Figure 2 cancers-14-00541-f002:**
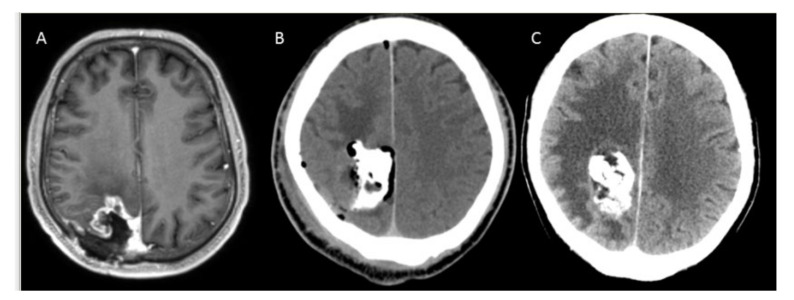
(**A**) Magnetic resonance imaging performed preoperatively showing contrast-enhanced tumor in the right parietal lobe. (**B**) Contrast-enhanced computed tomography performed at the first postoperative day. (**C**) Computed tomography in patient 13 demonstrating significant edema two months after surgery.

**Figure 3 cancers-14-00541-f003:**
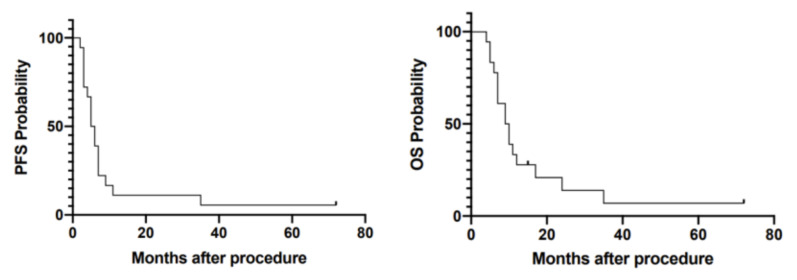
Kaplan–Meier plots demonstrating progression-free survival (PFS) and overall survival (OS) in months, after fluorescence-guided resection (FGR) and thermotherapy. Strokes mark censored data points. Median progression-free survival (PFS) for the study population was 5.5 months (95% CI: 4.67–6.13) after thermotherapy and median overall survival (OS) was 9.5 months (95% CI: 7.21–11.79).

**Figure 4 cancers-14-00541-f004:**
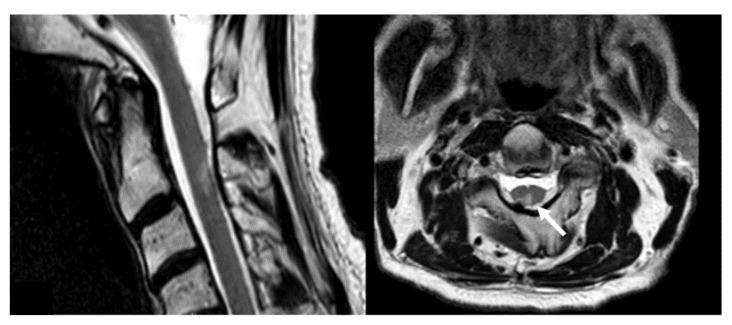
T2-weighted magnetic resonance imaging of the cervical spine of patient 4 in sagittal (**left**) and axial (**right**) planes. Note the intramedullary hyperintense lesion of the spinal cord at the level of C2-3 adjacent to metal artefacts (arrow).

**Figure 5 cancers-14-00541-f005:**
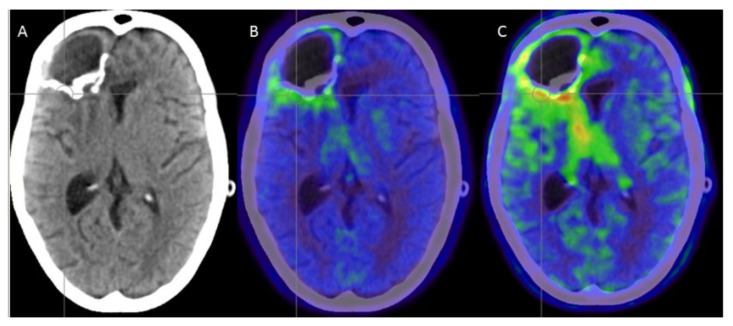
Patient 8, six months after completion of thermotherapy. Combination of CT post contrast (**A**), ^18^F-FET- (**B**) and ^18^F-DPA-714-PET (**C**) imaging. Note the increasing volumes of contrast enhancement, FET positivity and DPA positivity, respectively; the latter is referring to the infiltration of myeloid cells within the tumor microenvironment.

**Table 1 cancers-14-00541-t001:** Baseline patient characteristics.

ID	Age (Years)	Sex	Initial Diagnosis	Time Since Initial Diagnosis (Months)	Number of Recurrences	Current Diagnosis	MGMT Promotor	Location
1	42	F	Glioblastoma, IDH mutated	6	1	Glioblastoma, IDH mutated	methylated	LP
2	60	M	Glioblastoma, IDH wild type	13	2	Glioblastoma, IDH wild type	unmethylated	RP
3	59	M	Glioblastoma, IDH wild type	15	1	Glioblastoma, IDH wild type	unmethylated	LP
4	65	M	Glioblastoma, IDH wild type	8	2	Glioblastoma, IDH wild type	unmethylated	RP
5	75	F	Glioblastoma, IDH wild type	4	1	Glioblastoma, IDH wild type	unmethylated	RF
6	42	M	Glioblastoma, IDH wild type	3	4	Glioblastoma, IDH wild type	methylated	LF
7	63	F	Glioblastoma, IDH wild type	13	1	Glioblastoma, IDH wildtype	unmethylated	LP
8	36	F	Anaplastic astrocytoma, IDH mutated	57	2	Anaplastic astrocytoma, IDH mutated	unmethylated	RF
9	38	M	Glioblastoma, IDH wildtype	42	2	Glioblastoma, IDH wildtype	unmethylated	LO
10	62	M	Glioblastoma, IDH wildtype	7	1	Glioblastoma, IDH wildtype	unmethylated	LO
11	42	F	Glioblastoma, IDH wildtype	33	1	Glioblastoma, IDH wildtype	unmethylated	LO
12	40	F	Diffuse Astrocytoma, IDH mutated	248	3	Anaplastic Astrocytoma, IDH mutated	methylated	RF
13	58	M	Glioblastoma, IDH wildtype	6	1	Glioblastoma, IDH wildtype	unmethylated	RP
14	44	F	Glioblastoma, IDH wildtype	8	1	Glioblastoma, IDH wildtype	unmethylated	LF
15	46	M	Glioblastoma, IDH wildtype	10	1	Glioblastoma, IDH wildtype	unmethylated	RP
16	57	M	Diffuse Astrocytoma, IDH mutated	167	3	Anaplastic Oligodendroglioma, IDH mutated	not available	LT
17	58	F	Glioblastoma, IDH wildtype	16	3	Glioblastoma, IDH wildtype	unmethylated	LT
18	45	M	Glioblastoma, NOS	133	3	Glioblastoma, IDH mutated	methylated	LT

R, right. L, left. F, frontal. P, parietal. T, temporal. O, occipital.

**Table 2 cancers-14-00541-t002:** Data on thermotherapy and follow up.

Patient ID	EOR	Volume of Nanoparticles (mL)	Peak Temperature (°C)	Re-Irradiation (Gy)	Salvage Chemotherapy	Time to Edema (Months)	Revision Due to Edema	PFS (Months)	OS (Months)
1	STR	2.8	59	39.6	-	3.67	Yes	72 *	72 ^†^
2	GTR	3.2	60	39.6	-	0.33	No	7	7
3	NTR	2	56	39.6	-	1.97	Yes	11	24
4	STR	2.1	53	39.6	-	Non	Yes	6	9
5	GTR	5	53	-	-	4.43	No	5	7
6	NTR	3	54	-	-	Non	No	3	4
7	NRT	3.2	49	-	-	Non	No	3	5
8	GTR	4.3	44	-	Temozolomide	5.07	No	35	35
9	GTR	3.4	50	-	-	2.73	No	4	10
10	GTR	3.8	48	39.6	Bevacizumab	3.33	No	8	8
11	NTR	5	45	39.6	-	3.33	No	5	10
12	NTR	3	42	-		Non	No	7	17
13	NTR	7	46	39.6	-	1.5	Yes	6	7
14	GTR	3	39	39.6	-	2.0	No	5	6
15	GTR	1.9	43.3	39.6	-	2.33	Yes	3	15 ^†^
16	NTR	5	45	-	-	Non	No	7	12
17	STR	5	44.6	-	-	0.77	No	3	5
18	STR	5	45	-	-	1.83	Yes	2	11

EOR, extent of resection. GTR, gross total resection, NTR, near total resection. STR, subtotal resection. PFS, progression-free survival. OS, overall survival. * Patients without evidence of progression. † patients still alive.

## Data Availability

All data generated or analysed during this study are included in this published article.

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
