# Peer review of "Combined Fluorescence-Guided Resection and Intracavitary Thermotherapy with Superparamagnetic Iron-Oxide Nanoparticles for Recurrent High-Grade Glioma: Case Series with Emphasis on Complication Management"

_cancers, 2022, doi:10.3390/cancers14030541_

Round 1
Reviewer 1 Report
The manuscript by Dr. Schwake et al. reports a clinical case that high-grade glioma (HGG) can be treated with a combination of 5-ALA-fluorescence-guided resection and intracavitary thermotherapy with superparamagnetic iron-oxide nanoparticles. I think that this work is solid and interesting, especially with respect to the clinical trial of a new treatment option for local tumor control in HGG patients. The manuscript is clearly written and easy to understand. Therefore, I recommend this work for publication after minor revision. My comment is as follows:
1. What is the major cause of developing cerebral edema during the treatment?
2. Please suggest a feasible way to circumvent the side effect.
Author Response
We like to thank the reviewers for their time and their valuable comments.
Reviewer 1:
- What is the major cause of developing cerebral edema during the treatment?
Answer:
We believe that a major part of edema formation depends on immunological processes triggered by thermotherapy. As cerebral edema occurred both in patients with and without radiotherapy, a significant role of hyperthermia in edema development is to be assumed, although radiotherapy applied within a short time interval may still add to this process (Rubner et al. 2014).
Rubner Y, Muth C, Strnad A, Derer A, Sieber R, Buslei R, Frey B, Fietkau R, Gaipl US: Fractionated radiotherapy is the main stimulus for the induction of cell death and of Hsp70 release of p53 mutated glioblastoma cell lines. Radiat Oncol 9: 89, 2014.
Neurotoxicity directly attributed to re-irradiation is difficult to determine and has been found to be dose-dependent, but radiation-induced necrosis after normofractionated radiotherapy seems to be a rare event, and to develop as a late complication (Cho et al. 1999; Combs et al. 2005; 2008; Fetcko et al. 2017)
Cho KH, Hall WA, Gerbi BJ, Higgins PD, McGuire WA, Clark HB: Single dose versus fractionated stereotactic radiotherapy for recurrent high-grade gliomas. Int J Radiat Oncol Biol Phys 45: 1133-1141, 1999
Combs SE, Gutwein S, Thilmann C, Huber P, Debus J, Schulz-Ertner D: Stereotactically guided fractionated re-irradiation in recurrent glioblastoma multiforme. J Neurooncol 74: 167-171, 2005
Combs SE, Bischof M, Welzel T, Hof H, Oertel S, Debus J, Schulz-Ertner D: Radiochemotherapy with temozolomide as re-irradiation using high precision fractionated stereotactic radiotherapy (FSRT) in patients with recurrent gliomas. J Neurooncol 89: 205-210, 2008
Fetcko K, Lukas RV, Watson GA, Zhang L, Dey M: Survival and complications of stereotactic radiosurgery: A systematic review of stereotactic radiosurgery for newly diagnosed and recurrent high-grade gliomas. Medicine (Baltimore) 96: e8293, 2017).
Our previous investigations delivered several explanations for the generation of cerebral edema (Grauer et al. 2019):
Grauer O, Jaber M, Hess K, Weckesser M, Schmidt W, Maring S, Wölfer J, Stummer W. Combined intracavitary thermotherapy with iron oxide nanoparticles and radiotherapy as local treatment modality in recurrent glioblastoma patients. J Neurooncol. 2019 Jan; 141(1): 83-94.
We found massive tumor necrosis in the areas next to large nanoparticle aggregates. At the borders of the necrotic zone, apoptotic cell death, identified by Caspase-3 activation, and enhanced expression of HSP70 could be detected. We also found an increased infiltration of phagocytes with subsequent phagocytosis of necrotic debris and nanoparticles. Phagocytic cells also expressed myeloperoxidase, a well-known enzyme with strong pro-oxidative and proinflammatory properties. In addition, we observed enhanced infiltration by T-cells, NK-cells, tumor-associated microglia/macrophages. Profiling of these cells revealed a switch from an immunosuppressive Th2 to a pro-inflammatory Th1 phenotype.
These observations correlate closely with the temperature gradient induced by our approach: Temperature decreases from the inner to the outer zone of the treatment field with core temperatures above 45°C known to induce necrotic cell death and rim temperatures between 42 and 45 °C which usually cause apoptotic cell death. Temperatures above 40°C are necessary to stimulate the expression of HSPs which are known to be upregulated to protect cells against apoptosis (Dewey et al. 2008).
Dewey WC, Hopwood LE, Sapareto SA, Gerweck LE: Cellular responses to combinations of hyperthermia and radiation. Radiology 123: 463-474, 1977; Bettaieb A, Averill-Bates DA: Thermotolerance induced at a fever temperature of 40 degrees C protects cells against hyperthermia-induced apoptosis mediated by death receptor signalling. Biochem Cell Biol 86: 521-538, 2008
- Please suggest a feasible way to circumvent the side effect.
Answer:
We agree with the reviewer that further strategies must be developed to control edema formation after thermotherapy. As edema formation is probably an indicator for immunological effects at the treatment site (see above), administration of high doses of corticosteroids (dexamethasone more than 4 mg/day) for a longer period (> 6 weeks) should be avoided. As an alternative, bevacizumab could be used for edema control, as has been demonstrated for the treatment of radionecrosis (Levin et al. 2011). Otherwise, surgical removal of the nanoparticles and of necrotic tissue bulks from the tumor core must be discussed and is usually applied at the current state.
Levin et al. Randomized double-blind placebo-controlled trial of bevacizumab therapy for radiation necrosis of the central nervous system. Int. J. Radiation Oncology Biol. Phys., Vol. 79, No. 5, pp. 1487-1495, 2011.
Therefore, patient selection is critical. At the current state, thermotherapy should be limited to patients not in need of corticosteroids before treatment starts. In addition, patients with compromised clinical performance status who may not tolerate temporary increases in edema secondary to necrosis or are in risk of other major side effects will not be eligible for this therapy – as it is the case for all other therapies as well.
Moreover, there is room for improvement of technical aspects of the treatment:
As hyperthermia must be assumed as the major cause for edema formation, special attention has to be directed to temperature simulation and adjustment. We are currently working on solutions to measure temperatures simultaneously at multiple sites along the thermometry catheter, which should preferably be positioned along the rim of the tumor cavity. These measurements will be used for real time corrections of the temperature simulation to create a more precise temperature chart. The main determinants of tissue heat generation are field strength, nanoparticle density and tissue perfusion.
- Adjustment of the alternating magnetic field is possible within seconds.
- Three-dimensional nanoparticle distribution is stable over time and can be easily measured by CT.
- However, a standard perfusion value averaged over the whole brain is currently used for temperature simulation. Because of postoperative scarring, necrosis, varying cell densities or preexisting edema, etc., local perfusion will almost always differ from this generalized assumption. As brain perfusion mapping can be easily and routinely accomplished using dynamic contrast CT techniques, the NanoPlan software is currently modified to include this modality for a voxel-based correction of the simulation. The main determinants of tissue heat generation are nanoparticle density and tissue perfusion. Three-dimensional nanoparticle distribution is stable over time and directly influences temperature distribution by dictating the shape (but not the absolute temperature value) of the isothermic lines within the particle-instilled volume. Particle density is easily measured by CT, thus providing one necessary parameter for temperature simulation.
Accordingly, we have modified the complication management and outlook sections in the manuscript.
Reviewer 2 Report
Simple Summary: The authors performed a retrospective study on the combined use of fluorescence-guided surgery (via administering the prodrug 5-aminolevulinic acid) and intracavitary thermotherapy (via injecting superparamagnetic iron-oxide nanoparticles) on patients with high-grade gliomas. The authors found that the median PFS and median OS were 5.5 and 9.5 months, respectively.
Overall Impression: The use of these particular two modalities together for treating HGG is intriguing. However, due to reviewing cases from a single center, the retrospective study suffers from having a small sample size of patients with HGG, wherein the results might be otherwise (and even statistically significant) if the authors were able to use a larger sample size of patients.
- Background information on the PFS and OS of patients with HGG that are treated in accordance to typical standard of care (i.e., maximal resection with/without temozolomide and with/without radiotherapy) need to be provided such that a comparison of the techniques can be performed. That is, what are the median PFS and OS of patients with HGG that have undergone standard treatments?
- The is some confusion (ambiguity) in the manner that the authors wrote the "Results" section. The authors need to clearly identify groups or groups from within groups that received particular treatments. To do so, it would be great and I strongly suggest the authors clarify which groups (patients) received which treatment with the use of a branch diagram or some other visual aid to that effect.
- 5-ALA is notorious for providing inconsistent (both false-positives and false-negatives) results regarding the extent of its uptake and bioconversion into a fluorophore (i.e., protoporphyrin IX). Background information (PFS and OS) on this technique alone as well as using that of, say, an antibody-fluorophore construct (e.g., panitumumab/cetuximab-IRDye800) to the authors dual-mode review would be an interesting comparison, especially as the idea is to highlight that the dual-mode technique presented here could be beneficial (without regard to the edema issue). Could the authors comment on this in the background/introduction sections?
Author Response
- Background information on the PFS and OS of patients with HGG that are treated in accordance to typical standard of care (i.e., maximal resection with/without temozolomide and with/without radiotherapy) need to be provided such that a comparison of the techniques can be performed. That is, what are the median PFS and OS of patients with HGG that have undergone standard treatments?
Answer:
Standard care for the treatment of recurrent GBM is not defined. If patients are eligible, , re-resection combined with irradiation and or chemotherapy are usually offered. However, selection bias, the heterogeneity of the study populations, the lack of prospective randomized trials and different treatment regimes limit comparative conclusions.
For re-surgery groups have reported median PFS of 1.9-7 months and median OS of 6.5-12.9 months (Heleseth et al. 2010; Suchorska et al. 2016; Ringel et al. 2016).
Helseth, R.; Helseth, E.; Johannesen, T.B.; Langberg, C.W.; Lote, K.; Rønning, P.; Scheie, D.; Vik, A.; Meling, T.R. Overall survival, prognostic factors, and repeated surgery in a consecutive series of 516 patients with glioblastoma multiforme: Survival, prognostic factors, and repeat surgery in GBM patients. Acta Neurol. Scand. 2010, 122, 159–167.
Suchorska, B.; Weller, M.; Tabatabai, G.; Senft, C.; Hau, P.; Sabel, M.C.; Herrlinger, U.; Ketter, R.; Schlegel, U.; Marosi, C.; et al. Complete resection of contrast-enhancing tumor volume is associated with improved survival in recurrent glioblastoma-results from the DIRECTOR trial. Neuro-Oncology 2016, 18, 549–556.
Ringel, F.; Pape, H.; Sabel, M.; Krex, D.; Bock, H.C.; Misch, M.; Weyerbrock, A.; Westermaier, T.; Senft, C.; Schucht, P.; et al. Clinical benefit from resection of recurrent glioblastomas: Results of a multicenter study including 503 patients with recurrent glioblastomas undergoing surgical resection. Neuro-Oncology 2016, 18, 96–104.
Reirradiation in recurrent GBM was associated with a modest survival benefit compared to best supportive care alone. Survival times could be improved when re-irradiation was combined with other treatment modalities (from 8.2 mo up to 12.2 mo) (Shi et al 2018).
Shi, W.; Scannell Bryan, M.; Gilbert, M.R.; Mehta, M.P.; Blumenthal, D.T.; Brown, P.D.; Valeinis, E.; Hopkins, K.; Souhami, L.; Andrews, D.W.; et al. Investigating the effect of reirradiation or systemic therapy in patients with glioblastoma after tumor progression: A secondary analysis of NRG oncology/radiation therapy oncology group trial 0525. Int. J. Radiat. Oncol. Biol. Phys. 2018, 100, 38–44.
Regarding systemic treatments in recurrent GBM, therapeutic options include salvage chemotherapy with temozolomide, lomustine, bevacizumab or combinations, and regorafenib. Median PFS of 1.5 – 4.2 mo and median OS of 6.0 to 10.6 mo. were reported (Perry et al. 2010; Weller et al. 2015; Batchelor et al. 2013; Wick et al. 2017; Friedman et al. 2009; Lombardi et al. 2019).
Perry JR, Bélanger K, Mason WP, et al. Phase II trial of continuous dose-intense temozolomide in recurrent malignant glioma: RESCUE study. J Clin Oncol 2010a;28:2051-2057.
Weller M, Tabatabai G, Kastner B, et al. MGMT promoter methylation is a strong prognostic biomarker for benefit from dose-intensified temozolomide rechallenge in progressive glioblastoma: the DIRECTOR trial. Clin Cancer Res 2015; 21(9): 2057-64.
Batchelor TT, Mulholland P, Neyns B, et al. Phase III randomized trial comparing the efficacy of cediranib as monotherapy, and in combination with lomustine, versus lomustine alone in patients with recurrent glioblastoma. J Clin Oncol 2013; 31: 3212–8.
Wick, W.; Gorlia, T.; Bendszus, M.; Taphoorn, M.; Sahm, F.; Harting, I.; Brandes, A.A.; Taal, W.; Domont, J.; Idbaih, A.; et al. Lomustine and Bevacizumab in Progressive Glioblastoma. N. Engl. J. Med. 2017, 377, 1954–1963.
Friedman H, Prados M, Wen P, et al. Bevacizumab alone and in combination with irinotecan in recurrent glioblastoma. J Clin Oncol 2009;27:4733-4740.
Lombardi G, De Salvo GL, Brandes AA, Eoli M, Rudà R, Faedi M, Lolli I, Pace A, Daniele B, Pasqualetti F, Rizzato S, Bellu L, Pambuku A, Farina M, Magni G, Indraccolo S, Gardiman MP, Soffietti R, Zagonel V. Regorafenib compared with lomustine in patients with relapsed glioblastoma (REGOMA): a multicentre, open-label, randomised, controlled, phase 2 trial. Lancet Oncol. 2019 Jan;20(1):110-119. doi: 10.1016/S1470-2045(18)30675-2. Epub 2018 Dec 3. PMID: 30522967.
These informations were added to the manucript.
- There is some confusion (ambiguity) in the manner that the authors wrote the "Results" section. The authors need to clearly identify groups or groups from within groups that received particular treatments. To do so, it would be great, and I strongly suggest the authors clarify which groups (patients) received which treatment with the use of a branch diagram or some other visual aid to that effect.
Answer:
We added a flow chart to the manuscript. Surgery was performed, after informed consent of the patients, in cases where we assumed a complete or near to complete resection would be possible. Contraindication for surgery were unremovable metal implants close to surgical site (less than 40 cm) Moreover, patients with the necessity of opening the ventricle or basal cisterns during tumor resection were not eligible for application of nanoparticles to decrease risk of unintended particle distribution within the CSF spaces. All patients (n=18) were treated with nanoparticles. Re-irradiation was performed when possible (n=9). In cases with concomitant radiotherapy, the first thermotherapy session was scheduled 3 days before the start of radiotherapy, while another five sessions were conducted at days 1, 4, 8, 11, 15 ± 1 day. Radiotherapy took place within a time interval of 2 h to each thermotherapy session.
- 5-ALA is notorious for providing inconsistent (both false-positives and false-negatives) results regarding the extent of its uptake and bioconversion into a fluorophore (i.e., protoporphyrin IX). Background information (PFS and OS) on this technique alone as well as using that of, say, an antibody-fluorophore construct (e.g., panitumumab/cetuximab-IRDye800) to the authors dual-mode review would be an interesting comparison, especially as the idea is to highlight that the dual-mode technique presented here could be beneficial (without regard to the edema issue). Could the authors comment on this in the background/introduction sections?
Answer:
Fluorescence-guided resection with 5-aminolevulinic acid (5-ALA) is nowadays the only approved and one of the widely used procedures in adult glioma surgery. The efficacy of 5-ALA-assisted resection and its impact on the degree of resection, progression-free survival (PFS) and overall survival (OS) in primary cases has been demonstrated in numerous studies (Schucht et al. 2012; Stummer et al. 2006; Stummer et al. 2000; Díez Valle et al. 2014; Aldave et al. 2013). Stummer et al. identified a significantly higher OS when no residual tumor was visible after surgery (16.7 versus 11.8 mo, P < 0.0001) (Stummer et al. 2008). Moreover, this method is based on intracellular accumulation of fluorescent protoporphyrinIX (PpIX). This selective accumulation is the reason for the high sensitivity and specificity of the fluorescence signal (Valdés et al. 2010; Valdés et al. 2011; Stummer, Tonn, et al. 2014; Idoate et al. 2011; Hefti et al. 2008; Cage et al. 2013, Müther et al. 2019) also in case of recurrent malignant brain tumors (Nabavi et al. 2009).
Stummer W, Pichlmeier U, Meinel T, Wiestler OD, Zanella F, Reulen HJ; ALA-Glioma Study Group. Fluorescence-guided surgery with 5-aminolevulinic acid for resection of malignant glioma: a randomised controlled multicentre phase III trial. Lancet Oncol. 2006 May;7(5):392-401. doi: 10.1016/S1470-2045(06)70665-9. PMID: 16648043.
Stummer W, Novotny A, Stepp H, Goetz C, Bise K, Reulen HJ. Fluorescence-guided resection of glioblastoma multiforme by using 5-aminolevulinic acid-induced porphyrins: a prospective study in 52 consecutive patients. J Neurosurg. 2000 Dec;93(6):1003-13. doi: 10.3171/jns.2000.93.6.1003. PMID: 11117842.
Nabavi A, Thurm H, Zountsas B, Pietsch T, Lanfermann H, Pichlmeier U, Mehdorn M; 5-ALA Recurrent Glioma Study Group. Five-aminolevulinic acid for fluorescence-guided resection of recurrent malignant gliomas: a phase ii study. Neurosurgery. 2009 Dec;65(6):1070-6; discussion 1076-7. doi: 10.1227/01.NEU.0000360128.03597.C7. PMID: 19934966.
Díez Valle R, Slof J, Galván J, Arza C, Romariz C, Vidal C; VISIONA study researchers. Observational, retrospective study of the effectiveness of 5-aminolevulinic acid in malignant glioma surgery in Spain (The VISIONA study). Neurologia. 2014 Apr;29(3):131-8. English, Spanish. doi: 10.1016/j.nrl.2013.05.004. Epub 2013 Jul 17. PMID: 23870657.
Aldave G, Tejada S, Pay E, Marigil M, Bejarano B, Idoate MA, Díez-Valle R. Prognostic value of residual fluorescent tissue in glioblastoma patients after gross total resection in 5-aminolevulinic Acid-guided surgery. Neurosurgery. 2013 Jun;72(6):915-20; discussion 920-1. doi: 10.1227/NEU.0b013e31828c3974. PMID: 23685503.
Schucht P, Beck J, Abu-Isa J, Andereggen L, Murek M, Seidel K, Stieglitz L, Raabe A. Gross total resection rates in contemporary glioblastoma surgery: results of an institutional protocol combining 5-aminolevulinic acid intraoperative fluorescence imaging and brain mapping. Neurosurgery. 2012 Nov;71(5):927-35; discussion 935-6. doi: 10.1227/NEU.0b013e31826d1e6b. PMID: 22895402.
Valdés PA, Samkoe K, O'Hara JA, Roberts DW, Paulsen KD, Pogue BW. Deferoxamine iron chelation increases delta-aminolevulinic acid induced protoporphyrin IX in xenograft glioma model. Photochem Photobiol. 2010 Mar-Apr;86(2):471-5. doi: 10.1111/j.1751-1097.2009.00664.x. Epub 2009 Dec 7. PMID: 20003159; PMCID: PMC2875336.
Valdés PA, Leblond F, Kim A, Harris BT, Wilson BC, Fan X, Tosteson TD, Hartov A, Ji S, Erkmen K, Simmons NE, Paulsen KD, Roberts DW. Quantitative fluorescence in intracranial tumor: implications for ALA-induced PpIX as an intraoperative biomarker. J Neurosurg. 2011 Jul;115(1):11-7. doi: 10.3171/2011.2.JNS101451. Epub 2011 Mar 25. PMID: 21438658; PMCID: PMC3129387.
Stummer W, Tonn JC, Goetz C, Ullrich W, Stepp H, Bink A, Pietsch T, Pichlmeier U. 5-Aminolevulinic acid-derived tumor fluorescence: the diagnostic accuracy of visible fluorescence qualities as corroborated by spectrometry and histology and postoperative imaging. Neurosurgery. 2014 Mar;74(3):310-9; discussion 319-20. doi: 10.1227/NEU.0000000000000267. PMID: 24335821; PMCID: PMC4206350.
Piccirillo SG, Dietz S, Madhu B, Griffiths J, Price SJ, Collins VP, Watts C. Fluorescence-guided surgical sampling of glioblastoma identifies phenotypically distinct tumour-initiating cell populations in the tumour mass and margin. Br J Cancer. 2012 Jul 24;107(3):462-8. doi: 10.1038/bjc.2012.271. Epub 2012 Jun 21. PMID: 22722315; PMCID: PMC3405212.
Hefti M, von Campe G, Moschopulos M, Siegner A, Looser H, Landolt H. 5-aminolevulinic acid induced protoporphyrin IX fluorescence in high-grade glioma surgery: a one-year experience at a single institutuion. Swiss Med Wkly. 2008 Mar 22;138(11-12):180-5. PMID: 18363116.
Cage TA, Pekmezci M, Prados M, Berger MS. Subependymal spread of recurrent glioblastoma detected with the intraoperative use of 5-aminolevulinic acid: case report. J Neurosurg. 2013 Jun;118(6):1220-3. doi: 10.3171/2013.1.JNS121537. Epub 2013 Feb 19. PMID: 23421452.
Müther M, Koch R, Weckesser M, Sporns P, Schwindt W, Stummer W. 5-Aminolevulinic Acid Fluorescence-Guided Resection of 18F-FET-PET Positive Tumor Beyond Gadolinium Enhancing Tumor Improves Survival in Glioblastoma. Neurosurgery. 2019 Dec 1;85(6):E1020-E1029. doi: 10.1093/neuros/nyz199. PMID: 31215632; PMCID: PMC6855932.
More background informations on 5-ALA guided resection are now included into the introduction section.
Panitumumab/cetuximab-IRDye800 guided resection is an interesting approach showing promising results is some studies (Zhou et al. 2021, Miller et al. 2018) , but not approved yet, delivery and tumor uptake of antibody-fluorophore constructs as suggested might be a problem as well as inconsistent expression of the target structure (EGFR) throughout the tumor tissue.
Zhou Q, van den Berg NS, Rosenthal EL, Iv M, Zhang M, Vega Leonel JCM, Walters S, Nishio N, Granucci M, Raymundo R, Yi G, Vogel H, Cayrol R, Lee YJ, Lu G, Hom M, Kang W, Hayden Gephart M, Recht L, Nagpal S, Thomas R, Patel C, Grant GA, Li G. EGFR-targeted intraoperative fluorescence imaging detects high-grade glioma with panitumumab-IRDye800 in a phase 1 clinical trial. Theranostics. 2021 May 21;11(15):7130-7143. doi: 10.7150/thno.60582. PMID: 34158840; PMCID: PMC8210618.
Miller SE, Tummers WS, Teraphongphom N, van den Berg NS, Hasan A, Ertsey RD, Nagpal S, Recht LD, Plowey ED, Vogel H, Harsh GR, Grant GA, Li GH, Rosenthal EL. First-in-human intraoperative near-infrared fluorescence imaging of glioblastoma using cetuximab-IRDye800. J Neurooncol. 2018 Aug;139(1):135-143. doi: 10.1007/s11060-018-2854-0. Epub 2018 Apr 6. PMID: 29623552; PMCID: PMC6031450.
